# *Salvia* Spp. Essential Oils against the Arboviruses Vector *Aedes albopictus* (Diptera: Culicidae): Bioactivity, Composition, and Sensorial Profile—Stage 1

**DOI:** 10.3390/biology9080206

**Published:** 2020-08-04

**Authors:** Basma Najar, Luisa Pistelli, Francesca Venturi, Giuseppe Ferroni, Silvia Giovanelli, Claudio Cervelli, Stefano Bedini, Barbara Conti

**Affiliations:** 1Department of Pharmacy, University of Pisa, Via Bonanno 6, 56126 Pisa, Italy; basmanajar@hotmail.fr (B.N.); luisa.pistelli@unipi.it (L.P.); silvia.giovanelli84@gmail.com (S.G.); 2Centro Interdipartimentale di Ricerca Nutrafood “Nutraceutica e Alimentazione per la Salute”, Università di Pisa, via Bonanno 6, 56126 Pisa, Italy; francesca.venturi@unipi.it; 3Department of Agriculture, Food and Environment, via Del Borghetto 80, 56124 Pisa, Italy; Giuseppe.ferroni@unipi.it; 4CREA–Centro di Ricerca Orticoltura e Florovivaismo, Corso Inglesi 508, 18038 Sanremo, Italy; claudio.cervelli@crea.gov.it

**Keywords:** essential oil composition, insecticide, mosquitoes, mosquito-borne arboviruses diseases, repellent, sensory quality

## Abstract

Mosquito-borne arboviruses diseases cause a substantial public health burden within their expanding range. To date, their control relies on synthetic insecticides and repellents aimed to control the competent mosquito vectors. However, their use is hampered by their high economic, environmental, and human health impacts. Natural products may represent a valid eco-friendly alternative to chemical pesticides to control mosquitoes, and mosquito-borne parasitic diseases. The aim of this work was to combine the chemical and sensorial profiles with the bioactivity data of *Salvia* spp. essential oils (EOs) to select the most suitable EO to be used as a repellent and insecticide against the invasive mosquito *Aedes albopictus* (Diptera: Culicidae), vector of pathogens and parasites, and to describe the EOs smell profile. To do this, the EOs of four *Salvia* species, namely *S. dolomitica*, *S. dorisiana*, *S. sclarea*, and *S. somalensis* were extracted, chemically analyzed and tested for their bioactivity as larvicides and repellents against *Ae. albopictus*. Then, the smell profiles of the EOs were described by a panel of assessors. The LC_50_ of the EOs ranged from 71.08 to 559.77 μL L^−1^ for *S. dorisiana* and *S. sclarea*, respectively. *S. sclarea* EO showed the highest repellence among the tested EOs against *Ae. albopictus* females (RD_95_ = 12.65 nL cm^−2^), while the most long-lasting, at the dose of 20 nL cm^−2^, was *S. dorisiana* (Complete Protection Time = 43.28 ± 3.43 min). *S. sclarea* EO showed the best smell profile, while *S. dolomitica* EO the worst one with a high number of off-flavors. Overall, all the EOs, with the exception of the *S. dolomitica* one, were indicated as suitable for “environmental protection”, while *S. dorisiana* and *S. sclarea* were indicated as suitable also for “Body care”.

## 1. Introduction

Mosquitoes (Diptera: Culicidae) are among the most serious threats for humans because of their ability to transmit viruses and parasites. In particular, the arboviruses dengue, yellow fever, chikungunya, and Zika have recently expanded their geographical distributions and caused severe disease outbreaks in many urban populations [1,2,3]. Since the transmission of these viruses depends on the presence of the competent mosquito vectors *Aedes aegypti* and *Aedes albopictus* [3,4,5], measures such as repellent- or insecticide-treated nets, indoor spraying, or personal protection measures are needed to protect people from mosquito-borne infections [6]. To date, mosquitoes are mainly controlled by synthetic insecticides and repellents, but besides the quick development of resistance by insect pests, their use is often prohibitively expensive, unsustainable, and it poses relevant risks to humans and environmental health [7].

In this regard, essential oils (EOs) for their effectiveness, minimal toxicity to mammals, and low impact on the environment [8,9,10] have been recognized among the best alternative to synthetic chemicals. However, despite their insecticidal and repellent properties, EOs still do not have the expected broad use. In fact, besides the composition variability and the high volatility, their strong smell prevents their widespread application [11,12]. For these reasons, the acceptance of EOs to the human sensorial system is an important feature for their success as an ingredient in commercial products for topical use or for environmental protection. 

The genus *Salvia* comprises many easily cultivable species well-known in traditional medicine, all around the world [13]. Besides the medicinal effects, *Salvia* spp. EOs have also been extensively reported for their antimicrobial and antifungal activity [14,15]^,^ as well as for their repellent and insecticidal properties [16,17,18,19]. 

The aim of this work was, therefore, to evaluate the efficacy as repellents and insecticides of the EOs extracted from four cultivated *Salvia* species, namely *S. dolomitica* Codd, *S. dorisiana* Standl., *S. sclarea* L., and *S. somalensis* Vatke, against the filariasis vector *Aedes albopictus* Skuse (Diptera: Culicidae). Besides, we described the EOs smell profile for their possible use as active ingredients in the formulation of products for the environment or for personal use.

## 2. Materials and Methods

### 2.1. Plant Material, EOs Extraction and GC-MS Analysis

The EOs were obtained from air-dried aerial parts of *Salvia dolomitica* Codd, *Salvia dorisiana* Standl., *Salvia sclarea* L., and *Salvia somalensis* Vatke (Lamiaceae) plants cultivated at the Centro di Ricerca Orticoltura e Florovivaismo, (CREA) (Sanremo, Italy) and collected during the Summer 2017 (Appendix A). The EOs were obtained by extraction for 2 h in a Clevenger apparatus. After the hydro-distillation, the EOs were dehydrated using anhydrous sodium sulphate and stored at −4 °C until analysis. The EOs were chemically analyzed by gas chromatography-mass spectroscopy (GC–MS) by an Agilent 7890B gas chromatograph equipped with an Agilent HP-5MS capillary column (30 m × 0.25 mm; coating thickness 0.25 μm) and an Agilent 5977B single quadrupole mass detector (Agilent Technologies Inc., Santa Clara, CA, USA). Analytical conditions were as follows: carrier gas helium at 1 mL/min; injection of 1 μL (0.5% HPLC grade *n*-hexane solution); oven temperature programmed from 60 to 240 °C at 3 °C min^−1^; split ratio 1:25, injector and transfer line temperatures 220 and 240 °C, respectively. The parameters that were acquired were as follows: full scan, scan range of 30–300 m/z; scan time of 1 sec. The identification of the constituents was based on a comparison of the retention times with those of the authentic samples, comparing their linear retention indices relative to the series of *n*-hydrocarbons. Computer matching was also used against commercial (NIST 14 and ADAMS) and laboratory-developed mass spectra library built up from pure substances and components of known oils and Mass Spectra (MS) literature data [20,21,22,23,24].

### 2.2. Aedes albopictus Rearing

Adults of *Aedes albopictus* Skuse (Diptera: Culicidae) were obtained from eggs collected in the open field on Masonite strips put in black pots filled with 1 L of tap water. The strips were daily collected, transferred in 500 mL beakers, and submerged in tap water under room conditions (26 ± 2 °C; 60% relative humidity), photoperiod of 14:10 h (L:D), for eggs hatching. The emerged larvae were fed with cat food until pupation. The pupae (about 300 per cage) were put in cylindrical cages (Plexiglas, 35 d7 60 cm) with a front cotton access sleeve. The emerged adults were kept under room conditions and fed with sucrose solution (20%) [17,25]. 

### 2.3. EOs Larvicidal Activity 

Ten newly fourth-instar larvae (0-24 h) were put in a 250 mL beaker with 0.1% Tween 80 water solutions of the EOs. The EOs were tested at 50, 100, 150, 200, 300, 400, and 600 μL L^−1^. As a control, 10 larvae were put in 0.1% Tween 80 tap water solution. Four replicates for each treatment were performed. The mortality of the larvae was recorded after 24 h. During the tests, no food was given to the larvae [26]. Abbott’s formula [27] was used to adjust the mortality percentage rates of the treatments on the basis of the controls’ mortality.

### 2.4. Essential Oils Repellent Activity

The repellence of the EOs was evaluated by the human-bait technique [28] with some modifications. The experiments were performed during the summer in the above-described cages. The cages contained about of 300 8–12 day-old adults (sex ratio 1:1). The mosquitoes were starved for 12 h and were not blood-fed or exposed to any form of repellent. The tests were performed by ten volunteers not allergic to mosquito bites, and that had no contact with perfumed products on the day of the bioassay. All volunteers were informed about the experiment and provided their written consent. After rinsing the hands in distilled water, the volunteer’s forearms were protected with thick fabric sleeves and the hands with latex gloves in which a dorsal square area 5 × 5 cm was cut open. The mosquito-exposed skin of one hand was treated with 100 μL of ethanol as a negative control. The other hand was treated with 100 μL of EO ethanolic solution at concentrations ranging from 0.02 to 200 nL cm^−2^. After ethanol evaporation, the control hand was inserted inside the cage and exposed to mosquitoes for 3 min. Immediately after, the other hand was treated with the EO solution and, after ethanol evaporation, exposed to mosquitoes in the same cage. The number of probing mosquitoes was recorded by two observers. All the tests were performed between 8:00 and 10:00 am. The complete protection time (CPT) for the concentration of the EO of 0.2 µL cm^−2^ of skin was calculated by tests performed every 15 min until either two bites occurred in a single exposure period or one bite occurred in each of two consecutive exposure periods. The complete protection times were calculated on an average of six replicates. To verify the mosquitoes’ readiness to bite, the control and the EOs treated hand were regularly interchanged during each test. We considered the test valid if at least 30 mosquitoes landed on the control hand and attempted to bite. If the number of probing was < 30, a new mosquito’s cage was used [11]. The study was approved by the ethical committee of the University of Pisa (Comitato Bioetico dell’Università di Pisa).

### 2.5. Essential Oils Sensory Analysis

The sensory analysis of the EOs was performed by a panel of 10 assessors (four males and six females aged from 23 to 60 years) selected and trained for sensory analysis of foods (mainly wine, vegetal oils, and bakery products) and non-foods (mainly essential oils), according to the internal protocol of the Department of Agriculture, Food, and Environment (DAFE) of University of Pisa [29,30]. For this general training protocol, five training sessions specific for the assessment of *Sage* spp. were arranged until the assessors familiarized with the main descriptors useful for the characterization of aromatic plants. With this aim, during these training sessions, assessors were asked to identify and describe the smell profile of different solutions prepared by infusion (12 h, 25 °C of temperature, inert atmosphere) in hydroalcoholic solution (13% *v*/*v*) of the main aromatic plants, spices, different flowers, fruits, and fresh vegetables.

The smell assessment of the EOs tested was performed as a blinded test, in a quiet, well-ventilated room in the morning. Each assessor was provided with a filter paper (2 × 2 cm) soaked with 20 µL of EO (1%). To avoid cross-contamination, the five samples were assessed separately in the same panel session (15 min waiting between two tests). The assessors were provided with a specific non-structured parametric descriptive scoring chart, and described the main odors of each sample on the basis of descriptors ranked on a scale of 0–10 in terms of “Smell intensity”, “Smell persistency”, and, “Overall pleasantness” as hedonic parameters [31]. The assessors also evaluated the possible use (Body care or Environmental protection) of the EOs, as well as the main emotions (Familiar, Relax, Exotic, Repulsion) elicited by them.

### 2.6. Statistical Analysis

Data of the smell profiles assessments of EOs were analyzed by one-way ANOVA with the EO as factor. Equality of variances was checked before the analyses by the Levene’s test. The averages were separated by the Tukey’s b post-hoc test. Larvae median and lethal concentration to 95% of tested organisms (LC_50_, LC_95_), and adult median and total repellent dose (RD_50_, RD_95_) were calculated by probit regression. Differences between LC_50_, LC_95_, RD_50_, and RD_95_ values were evaluated by the relative median potency (rmp). The complete protection time (CPT) data were processed by the Kruskal–Wallis test with the time of protection as a factor. The means were separated by Dunn-Bonferroni pairwise comparisons. All the analyses were performed by the SPSS 22.0 software (SPSS Inc., Chicago, IL, USA).

## 3. Results

### 3.1. Essential Oils Chemical Composition

In total, 108 compounds were identified in the four *Salvia* spp. EOs, with an identification percentage ranging from 96.9% to 99.8% (Table 1).

The results showed that almost all the EOs were characterized by the total monoterpenes (from 52.84 to 90.87% in *S. dorisiana* and *S. sclarea*, respectively) as the main class of constituents, even though differently distributed among hydrocarbons and oxygenated derivatives. *Salvia sclarea* exhibited the highest percentage of oxygenated monoterpenes (68.46%), while *S. dorisiana* and *S. somalensis* pointed out nearly a similar amount in oxygenated monoterpenes (Table 1). The highest percentage of non-terpene derivative (20.66%) was detected in the *S. dorisiana* EO. On the other hand, *S. dolomitica* evidenced the highest percentage of sesquiterpenoids (51.06% of sesquiterpene hydrocarbons and 12.33% of oxygenated sesquiterpenes). Regarding the main identified constituents, both bornyl acetate and camphor were the main constituents of *S. somalensis* EO (18.10% and 12.91%, respectively) (Table 1). This latter species also evidenced an interesting amount of α-pinene (6.77%), camphene (6.05%), δ-3-carene (5.19%), β-caryophyllene (3.62%), *τ*-cadinol (5.12%), and limonene (4.13%).

*Salvia dolomitica* was characterized by the highest amount of β-caryophyllene (14.81%) followed by eucalyptol (10.17%) and aromadendrene (7.96%). δ-cadinene, borneol, γ-cadinene and, viridiflorene showed also a good relative percentage (5.86 > 4.41 > 4.36 > 3.96%, respectively), and β-eudesmene and α-eudesmol were exclusive constituents of this EO.

*Salvia dorisiana* EO showed a different composition, in fact, it was rich in perillyl acetate (21.74%) and methyl perillate (19.16%), together with β-caryophyllene (9.99%) and myrtenyl acetate (4.03%). It is interesting to note that perillyl acetate, methyl perillate, and myrtenyl acetate were present only in *S. dorisiana* EO, Linalyl acetate (32.03%) and α-thujene (9.91%) characterized *S. sclarea* EO, followed by linalool (11.90%).

### 3.2. Essential Oils Larvicidal Activity

The *Salvia* EOs showed a wide range of toxicity against the Asian tiger mosquito larvae, depending on the species. Overall, the toxicity of *S. dorisiana* EO (LC_50_ = 71.08 μL L^-1^) was in line with the toxicity previously reported for other aromatic plants’ EOs (LC_50_ ranging from 35 to 194 μL L^-1^) (reviewed by Pavela, 2015), while a much lower toxicity against *Ae. albopictus* larvae was recorded for *S. dolomitica*, *S. dorisiana*, and *S. somalensis*. (LC_50_ ranging from 315.52 to 559.77 μL L^-1^) (Table 2).

The comparison of the relative toxicity of *Salvia* spp. EOs against *Ae. albopictus* by rmp analysis of probits showed that the *S. dorisiana* EO was significantly more toxic to the mosquitoes’ larvae than the other *Salvia* EOs. In addition, *S. dolomitica* was significantly more toxic than *S. sclarea* EO (Table 3).

### 3.3. Essential Oils Repellent Activity

All the *Salvia* spp. EOs showed a repellent activity against the *Ae. albopictus* adults. RD_50_ values ranged from 0.56 to 5.03 nL cm^−2^ for *S. dorisiana* and *S. somalensis*, respectively while RD_95_ values ranged from 12.65 to 8308.54 nL cm^−2^ for *S. sclarea* and *S. somalensis*, respectively (Table 4).

The comparison of the relative toxicity of *Salvia* spp. EOs by rmp analyses did not show significant differences among the EOs, with the exception of *S. somalensis* EO that was significantly less repellent than the others (Table 5).

The results of the complete protection time (CPT) assay indicated that the *Salvia* EOs applied to human skin at the dose of 20 nL cm^−2^ protect from the mosquito’s bites for a time ranging from 4.60 ± 2.7 to 43.28 ± 3.43 min (for *S. somalensis* and *S. dorisiana*, respectively) (Table 6). The Kruskas–Wallis test indicated significant differences in the duration of the protection by *Salvia* EOs (*χ2* = 14.432; df = 4; *P* = 0.006). The Dunn–Bonferroni pairwise comparisons of the CPT values indicated that the protective effect of *S. dorisiana* was significantly longer-lasting than the one of the *S. somalensis* EO (Table 6).

### 3.4. Essential Oils Sensory Analysis

The panel tests indicated *S. dorisiana* as the EO with the highest “Smell intensity” (Figure 1), closely followed by *S. dolomitica* and *S. somalensis* as the EO with the lowest intensity. Such differences, however, were not statistically significant (*F*_(4.25)_ = 2.12. *p* = 0.109). On the contrary, the assessors attributed to the different EOs a significantly different “Smell persistency” as well as “Overall pleasantness” (*F*_(4.25)_ = 5.80, *p* = 0.002; *F*_(4.25)_ = 10.83, *p* < 0.001, respectively).

As for smell persistency, the highest value was attributed to *S. dorisiana*, followed by *S. dolomitica* and *S. sclarea* EO, with the lowest values attributed to the *S. somalensis* EO (Figure 1).

The best smell profile (highest value of overall pleasantness) was attributed to the *S. sclarea* EO, followed by the *S. somalensis* and the *S. dorisiana* EOs. The worst smell profile was attributed to the *S. dolomitica* EO that was characterized by a high number of off-flavors (Table 7).

Furthermore, *S. sclarea* and *S. somalensis* EOs showed the highest complexity in flavor descriptors (Table 7) referring to both family of vegetative and Spicy odors.

All the assessors associated the *S. dolomitica* EO to negative emotion “Repulsion”. On the contrary, the other *Salvia* spp. EOs were associated with positive sensations “Familiar” and “Relax” together with an “Exotic” feeling.

As for the possible use of the EOs, the assessors indicated that all the *Salvia* spp. EOs, with the exception of *S. dolomitica*, were suitable for “Environmental protection”, while only the *S. dorisiana* and *S. sclarea* were suitable also for “Body care” (Figure 2).

## 4. Discussion

The use of EOs as insecticides and insect repellents is raising increasing interest for their low impact on environmental and human health, and for their perception by consumers as safe, natural products [32]. As for their use as repellents, however, in order for them to be successfully utilized as an ingredient in commercial products for topical or for environmental protection use, the EOs need to have a good level of acceptance to the human sensorial system. In this work, we observed that the *Salvia* EOs tested showed a wide range of toxicity against the Asian tiger mosquito larvae, depending on the *Salvia* species and were able to show a repellent activity against the *Ae. albopictus* adults, depending on the concentration, while not all of them were indicated by the panelists as suitable to be utilized in the formulation of products for environmental protection or personal protection.

In order to get insights about the molecular components of the tested EOs that may act as chemical cues for mosquitoes and the human sensorial system, their chemical composition was characterized by GC-EIMS. The EO profiles of the studied species were completely in agreement with what was previously reported on *S. somalensis* [33] and *S. dorisiana* [17]. The *S. sclarea* EO evidenced results in partial agreement with those observed by Aćimović and co-workers [34] on clary sage EO from Tajikistan. The *S. dolomitica* EO was a subject of discord, in fact, Kamatou et al. [35] and Bassolino et al. [36] found a composition similar to what we found in the current study, while in a more recent study by Ebani [33], eucalyptol was the main compound. Caser underlined a completely different profile where limonene (19.8%), δ-3-Carene (9.1%), and germacrene D (8.6%) dominated together with (E)-β-ocimene (7.39%) and β-caryophyllene (7.9%) [37].

Overall, the toxicity of *S. dorisiana* EO against the *Ae. albopictus* larvae recorded in this work (LC_50_ = 71.08 μL L^−1^) agrees with previous studies performed with other aromatic plants EOs (LC_50_ ranging from 35 to 194 μL L^−1^) (reviewed by Pavela [38]) and was shown to be significantly higher than the other *Salvia* spp. EOs. In line with our findings, a strong variability in the larval toxicity of *Salvia* spp. EOs was observed also by Ali [16] against the mosquitoes *Anopheles quadrimaculatus* and *Aedes aegypti* (Diptera: Culicidae). These results indicate a strong variability in the toxicity of the EOs among species of the genus *Salvia* that is consistent with the high variability in the chemical composition of the EOs.

Besides the high variability in the chemical composition of the EOs, a limitation in the use of the EOs as mosquito repellent is represented by their high volatility. This is why the determination of the protection time is an important parameter for the screening of EOs for possible practical use. In this experiment, *S. dorisiana* EO showed a very good persistency (CPT = 43 min at the dose of 20 nL cm^−2^ of skin) of the repellent effect that was longer than the one reported for the synthetic repellent *N*,*N*-Diethyl-*meta*-toluamide (DEET) (CPT = 10 min at the dose of 40 nL cm^−2^ of skin) [11]. Overall, *S. dorisiana* showed the highest toxicity against *Ae. albopictus* larvae coupled with the strongest protection effect against the *Ae. albopictus* bites that were about 60% longer than that of DEET. On the contrary, *S. somalensis*, was the least toxic and persistent EO among the *Salvia* species tested.

These results are in line with previous experiments by Conti et al. [17] who found that *S. dorisiana* and *S. sclarea* EOs were able to protect the human skin from *Ae. albopictus* bites up to 31 and 21 min (at 40 nL cm^−2^), respectively. The repellent effect showed by *Salvia* spp. EOs in this work is consistent with one of EOs extracted from aromatic plants belonging to other families. For example, the EOs extracted from *Curcuma longa* L. (Zingiberaceae), *Pogostemon heyneanus* Benth. (Lamiaceae), and *Zanthoxylum limonella* Alston (Rutaceae) were found to repel the Asian tiger mosquito up to 23 min at a dose of 5% (about 10 times more concentrated than the solutions tested in the present study) [39]. Moreover, Nasir [40] found that the *Zingiber officinale* Rosc. (Zingiberaceae) *Mentha piperita* L., and *Ocimum basilicum* L. (Lamiaceae) showed 34–98 min of protection (EOs, 10%).

The sensorial analyses of the *Salvia* spp. EOs indicated that, overall, the best smell profile (highest value of overall pleasantness) was the one of the *S. sclarea* EO, followed by the *S. somalensis* and *S. dorisiana* EOs. The preference showed by the assessors to the *S. sclarea* EO should be probably due to the green spicy, fresh mint, menthol-like and citrus-like fragrances detected in all the samples. Those fragrances may be associated with the presence of the alcoholic volatile compounds’ linalool and *p*-cymen-8-ol (Table 1).

On the contrary, the worst smell profile attributed to *S. dolomitica* EO is probably due to the off-flavors detected by panelists in EO that can be well explained by the presence in volatile fraction of borneol (smell character: camphoraceous odor [41]) in association with β-caryophyllene (smell character: dry-woody-spicy, clove-like odor [41]) and β-pinene (smell character: dry-woody, resinous odor [41]).

## 5. Conclusions

The control of mosquitoes is paramount in the efforts of stopping the spread of mosquito-borne diseases. According to our results, *Salvia* species may represent a valid source of repellents and insecticides alternative to the synthetic ones for the control of mosquitoes. In fact, unlike synthetic chemicals, the low-cost and the increasing demand for effective and safe natural products may make EO-based products well accepted by consumers, provided that they are compatible with their sensory system. The multidisciplinary approach of this study showed that the selection of EOs as ingredients for effective and good-smelling anti-mosquito formulations is feasible by combining the data of the biological activity with their chemical and smell profiles.

## Figures and Tables

**Figure 1 biology-09-00206-f001:**
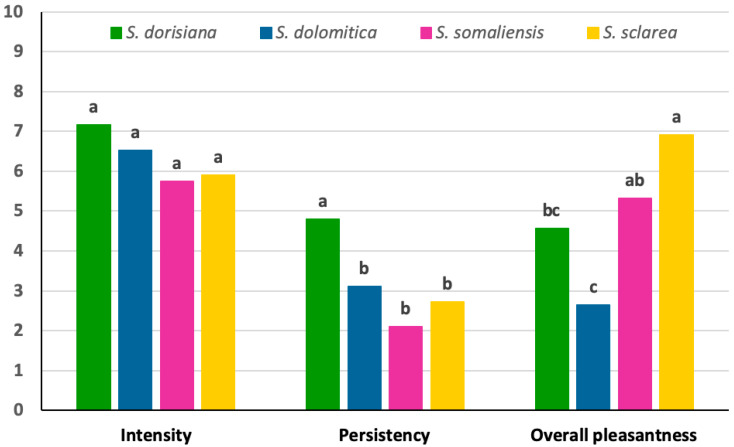
Smell characterization of the *Salvia dorisiana*, *S. dolomitica*, *S. sclarea*, and *S. somalensis* essential oils.

**Figure 2 biology-09-00206-f002:**
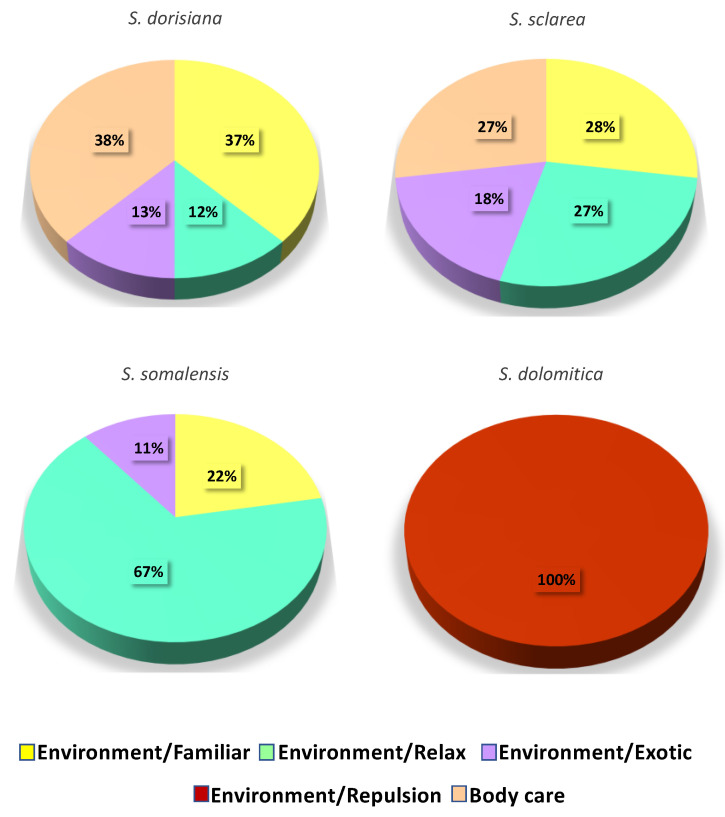
Main possible uses and emotions associated with the *Salvia dorisiana*, *S. dolomitica*, *S. sclarea*, and *S. somalensis* essential oils.

**Table 1 biology-09-00206-t001:** Chemical composition of *Salvia dolomitica*, *S. dorisiana*, *S. sclarea*, and *S. somaliensis* essential oils.

N°	Compounds	Rt		Class	LRI ^1^	LRI ^2^		*S. dolomitica*	*S. dorisiana*	*S. sclarea*	*S. somalensis*
								Relative Percentage (%)			
1	Tricyclene	5.28		MH	925	921		-	-	0.10	0.29
2	α-Thujene	5.36		MH	930	924		-	-	9.91	-
3	α-Pinene	5.55		MH	937	932		2.22	2.53	3.87	6.77
4	Camphene	5.94		MH	952	946		0.65	1.24	1.79	6.05
5	Sabinene	6.58		MH	978	968		0.21	-	0.10	-
6	β-Pinene	6.69		MH	979	974		1.23	0.56	2.86	0.75
7	Myrcene	7.05		MH	991	988		1.72	1.19	0.40	1.07
8	α-Phellandrene	7.49		MH	1005	1002		0.33	0.32	-	0.48
9	δ-3-carene	7.68		MH	1011	1008		3.28	2.16	-	5.19
10	1,4-Cineole	7.81		OM	1016	1012		-	-	0.49	-
11	α-Terpinene	7.88		MH	1017	1014		0.14	0.27	0.22	0.45
12	*o*-Cymene	8.00		MH	1022	1020		2.02	0.98	-	0.10
13	*p*-Cymene	8.14		MH	1025	1022		-	-	1.65	1.76
14	Limonene	8.28		MH	1030	1024		3.81	3.70	1.34	4.13
15	Eucalyptol	8.37		OM	1032	1026		10.17	5.83	4.96	-
16	*cis*-β-Ocimene	8.55		OM	1038	1032		1.58	0.87	-	-
17	*trans*-β-Ocimene	8.91		OM	1049	1044		0.16	0.23	0.10	-
18	γ-Terpinene	9.31		MH	1060	1054		0.26	0.49	0.17	0.53
19	*cis*-4-Thujanol	9.56		OM	1070	1074 ^$^		0.11	-	-	-
20	*cis*-Linalool oxide (furanoid)	9.79		OM	1074	1067		-	-	0.10	-
21	2-methoxyethyl-Benzene	10.27		NT	1087	1080		0.15	-	-	-
22	Terpinolene	10.38		MH	1088	1086		-	0.29	-	0.84
23	Fenchone	10.40		OM	1096	1083		0.45	-	0.45	-
24	Linalool	10.78		OM	1099	1095		-	0.23	11.9	0.21
25	*iso*-Amyl 2-methyl butyrate	11.03		NT	1101	1100 ^$^		-	0.26	-	-
26	β-Thujone	11.45		OM	1114	1112		-	-	2.40	-
27	*trans*-Sabinol	12.35		OM	1143	1137		0.39	-	-	-
28	Camphor	12.55		OM	1145	1141		0.27	-	8.10	12.91
29	Borneol	13.38		OM	1167	1165		4.41	3.61	1.08	3.35
30	4-Terpineol	13.86		OM	1177	1174		0.61	0.49	0.20	0.79
31	*p*-Cymen-8-ol	14.16		OM	1183	1179		-	-	-	0.17
32	α-Terpineol	14.14		OM	1189	1186		0.41	0.48	2.93	1.87
33	Myrtenol	14.64		OM	1195	1194		-	0.49	-	-
34	γ-Terpineol	14.68		OM	1197	1199		-	-	0.18	-
35	Nerol	15.97		OM	1228	1227		-	-	0.22	-
36	Linalyl acetate	17.18		OM	1257	1254		-	-	32.03	-
37	Bornyl acetate	18.41		OM	1285	1284		-	-	0.84	18.1
38	*p*-Mentha-1,8-dien-7-ol	18.89		OM	1296	1297 ^$^		-	1.11	-	-
39	Carvacrol	19.07		OM	1299	1298		-	-	0.47	-
40	Myrtenyl acetate	20.06		OM	1327	1324		-	4.03	-	-
41	α-Cubebene	21.06		SH	1351	1345		0.76	-	-	0.22
42	Eugenol	21.36		OM	1357	1356		-	-	0.23	-
43	Neryl acetate	21.71		OM	1364	1359		-	-	0.74	-
44	Ylangene	21.96		SH	1372	1373		-	-	-	0.52
45	Isoledene	22.03		SH	1375	1374		0.5	-	-	-
46	α-Copaene	22.15		SH	1376	1374		2.76	0.82	0.27	2.59
47	Geranyl acetate	22.51		OM	1382	1379		-	-	1.04	-
48	β-Cubebene	22.74		SH	1389	1387		0.10	-	-	-
49	*cis*-Jasmone	23.02		NT	1393	1392		-	-	-	0.37
50	Methyl perillate	23.23		NT	1394	1392		-	19.16	-	-
51	β-Panasinsene	23.33		SH	1395	1381		0.17	-	-	-
52	β-Maaliene	23.45		SH	1405	1411 ^$^		-	-	-	0.57
53	α-Gurjunene	23.53		SH	1410	1409		1.17	-	-	-
54	β-Caryophyllene	23.92		SH	1419	1417		14.81	9.99	3.47	3.62
55	β-Gurjunene	24.27		SH	1432	1431		1.46	-	-	0.15
56	1,1,3a-Trimethyl-7-methylenedecahydro-1H-cyclopropa[a]naphthalene	24.56		SH	1434	1435		-	0.18	-	-
57	*trans*-α-Bergamotene	24.60		SH	1435	1432		-	-	0.12	-
58	*p*-Mentha-1,8-dien-7-yl acetate	24.69		OM	1436	1436 ^$^		-	21.74	-	-
59	Aromadendrene	24.84		SH	1440	1439		7.96	-	-	1.00
60	α-Maaliene	24.89		SH	1443	1442 ^$^		0.96	-	-	0.12
61	Selina-5,11-diene	25..12		SH	1447	1447 ^$^		0.95	0.15	-	-
62	α-Humulene	25.29		SH	1454	1452		1.57	0.76	2.22	0.27
63	Cadina-3,5-diene	25.32		SH	1458	1454 ^$^		-	-	-	0.45
64	Alloaromadendrene	25.58		SH	1461	1458		0.87	0.19	-	0.33
65	γ-Muurolene	26.24		SH	1477	1478		0.72	0.17	-	1.10
66	α-Amorphene	26.32		SH	1482	1483		-	-	-	0.17
67	Germacrene D	26.40		SH	1485	1484		-	-	0.10	-
68	β-Eudesmene	26.51		SH	1486	1485 ^$^		1.02	-	-	-
69	Phenethyl isovalerate	26.79		NT	1490	1491 ^$^		-	0.18	-	-
70	δ-Selinene	26.83		SH	1493	1492		0.27	-	-	-
71	*epi*-Bicyclosesquiphellandrene	26.85		SH	1494	1490 ^$^		-	-	-	0.42
72	Viridiflorene	27.00		SH	1497	1496		3.96	0.75	0.10	-
73	Eremophilene	27.07		SH	1498	1498 ^$^		-	-	-	0.63
74	α-Muurolene	27.11		SH	1499	1500		0.53	-	-	0.99
75	β-Bisabolene	27.52		SH	1509	1505		-	-	0.10	-
76	γ-Cadinene	27.81		SH	1513	1513		4.36	1.04	-	2.67
77	δ-Cadinene	28.09		SH	1524	1522		5.86	2.18	0.11	5.66
78	Cubenene	28.48		SH	1532	1522 ^$^		0.12	-	-	0.48
79	α-Cadinene	28.61		SH	1538	1537		0.18	-	-	0.17
80	α-Calacorene	28.81		SH	1542	1544		-	0.13	-	0.55
81	Myrtenyl 2-methyl butyrate	29.55		NT	1560	1559 ^$^		-	0.20	-	-
82	(*E*)-Nerolidol	29.64		OS	1563	1561		-	0.27	-	1.49
83	Spathulenol	30.12		OS	1576	1577		0.50	0.13	-	-
84	Globulol	30.36		OS	1580	1590		3.36	-	-	0.57
85	Caryophyllene oxide	30.39		OS	1583	1582		-	1.62	0.94	-
86	Viridiflorol	30.71		OS	1591	1592		0.25	-	0.67	0.37
87	Ledol	30.99		OS	1599	1602		0.37	-	-	-
88	Rosifoliol	31.12		OS	1600	1600		1.55	-	-	-
89	Humulene epoxide II	31.31		OS	1606	1608		0.17	0.10	0.26	-
90	Di-*epi*-1,10-cubenol	31.54		OS	1614	1618		0.18	-	-	0.18
91	Junenol	30.77		OS	1617	1618		-	0.53	-	1.36
92	(*E*)-Farnesene epoxide	30.89		OS	1624	1624 ^$^		-	0.35	-	-
93	Epicubenol	31.00		OS	1627	1627		0.63	-	-	-
94	γ-Eudesmol	31.16		OS	1632	1630		0.33	-	-	-
95	*τ*-Cadinol	32.52		OS	1640	1638		2.18	2.42	-	5.12
96	δ-Cadinol	32.66		OS	1645	1646 ^$^		0.19	-	-	-
97	octahydro-2,2,4,7a-tetramethyl-1,3a-ethano(1H)inden-4-ol	32.78		OS	1648	1648 ^$^		-	0.33	-	-
98	α-Cadinol	32.93		OS	1653	1652		-	0.17	-	0.28
99	α-Eudesmol	32.99		OS	1655	1652		2.48	-	-	-
100	Aromadendrene oxide-(2)	33.97		OS	1678	1678 ^$^		-	0.16	-	-
101	α-Bisabolol	34.12		OS	1684	1683		-	0.26	-	-
102	Shyobunol	34.56		OS	1701	1686 ^§^		0.14	0.25	-	-
103	Farnesyl acetone	42.13		AC	1919	1913		-	0.11	-	-
104	Cembrene	43.63		DH	1939	1937		-	0.23	-	-
105	(*E*)-1-(6,10-Dimethylundec-5-en-2-yl)-4-methylbenzene	44.02		NT	1951	1950 ^$^		-	0.86	-	-
106	Gerany-*p*-cymene	44.98		DH	1980	1980 ^$^		-	0.11	-	-
107	*epi*-13-Manool	48.77		OD	2056	2059		-	-	0.10	-
108	Sclareol	51.14		OD	2227	2222		-	-	0.43	-
	Class of compounds							*S. dolomitica*	*S. dorisiana*	*S. sclarea*	*S. somalensis*
	Monoterpene Hydrocarbons (MH)							15.87	13.73	22.41	28.41
	Oxygenated Monoterpenes (OM)							18.56	39.11	68.46	37.40
	Total monoterpenes							34.43	52.84	90.87	65.81
	Sesquiterpene Hydrocarbons (SH)							51.06	16.36	6.49	22.68
	Oxygenated Sesquiterpenes (OS)							12.33	6.59	1.87	9.37
	Total sesquiterpenes							63.39	22.95	8.36	32.05
	Diterpene Hydrocarbons (DH)							-	0.34	-	-
	Oxygenated Diterpenes (OD)							-	-	0.53	-
	Apocarotenoids (AC)							-	0.11	-	-
	Non-terpene Derivatives (NT)							0.15	20.66	-	0.37
	Total Identified							97.97	96.90	99.76	98.23

LRI ^1^: Linear retention indices on DB-5 column. LRI^2^: Linear retention indices reported by Adams 1995; ^$^: Linear retention indices in NIST 2014 (https://webbook.nist.gov/chemistry/name-ser/); ^§^: Linear retention indices in pherobase (www.pherobase.com/database/kovats).

**Table 2 biology-09-00206-t002:** Toxicity of *Salvia dorisiana*, *S. dolomitica*, *S. sclarea*, and *S. somalensis* essential oils (EOs) against larvae of *Aedes albopictus*.

EO	LC_50_ ^a^	LC_95_ ^b^	*χ2* (df)	*P*
*S. dolomitica*	315.52 (293.24–338.14)	503.04 (454.85–582.09)	3.50 (8)	0.899
*S. dorisiana*	71.08 (65.91–76.14)	125.52 (112.70–146.54)	4.83 (8)	0.775
*S. sclarea*	559.77 (470.17–718.35)	2159.94 (1457.00–3974.90)	5.68 (8)	0.683
*S. somalensis*	388.51 (356.59–430.74)	686.63 (581.29–912.44)	0.99 (8)	0.998

^a^, Concentration of the EO that kills 50% of the exposed larvae; ^b^, concentration of the EO that kills 95% of the exposed larvae. Data are expressed as μL L^-1^; in bracket, confidence interval; df, degrees of freedom. *P*, significance level of Pearson goodness of fit test.

**Table 3 biology-09-00206-t003:** Relative toxicity of *Salvia dorisiana*, *S. dolomitica*, *S. sclarea*., and *S. somalensis* essential oils (EOs) against *Aedes albopictus* larvae.

	EO (X)	*S. dolomitica*	*S. dorisiana*	*S. sclarea*	*S. somalensis*
EO (Y)					
*S. dolomitica*		**-**	**5.48(3.00–13.24)**	**0.69 (0.45–1.01)**	0.73 (0.46–1.12)
*S. dorisiana*		**0.18 (0.08–0.33)**	-	**0.13 (0.05–0.25)**	**0.13 (0.05–0.27)**
*S. sclarea*		**1.46 (1.01–2.23)**	**7.98(4.07–21.81)**	-	1.07 (0.71–1.62)
*S. somalensis*		1.37 (0.90–2.18)	**7.49(3.77–20.47)**	0.94 (0.62–1.42)	-

Relative median potency analyses (rmp) values of probits (EO in column vs. EO in row): Values < 1 indicate higher repellence. Values > 1 indicate lower repellence. Bold indicates significant values (95% CI ≠ 1).

**Table 4 biology-09-00206-t004:** Repellence of *Salvia dorisiana*, *S. dolomitica*, *S. sclarea*, and *S. somalensis* essential oils (EOs) against *Aedes albopictus* females.

EO	RD_50_ ^a^	RD_95_ ^b^	*χ*2 (df)	*P*
*S. dolomitica*	0.98 (0.62–1.54)	38.68 (18.50–86.26)	66.35 (7)	<0.001
*S. dorisiana*	0.56 (0.19–1.11)	39.88 (7.84–46986.17)	12.59 (3)	0.006
*S. sclarea*	1.13 (0.74–1.74)	12.65 (6.12–52.92)	28.36 (4)	<0.001
*S. somalensis*	5.03 (3.69–7.01)	8308.54 (3387–25371.72)	14.91 (8)	0.061

^a^, dose of EO that reduces the number of landings to 50%; ^b^, dose of EO that reduces the number of landings to 95%. Data are expressed as nL cm^−2^ of skin. In bracket, confidence limits. *P*, significance level of Pearson goodness of fit test. Since *P* < 0.150, a heterogeneity factor is used in the calculation of confidence limits.

**Table 5 biology-09-00206-t005:** Relative repellence of *Salvia dorisiana*, *S. dolomitica*, *S. sclarea,* and *S. somalensis* essential oils (EOs) against *Aedes albopictus* females.

	EO (X)	*S. dolomitica*	*S. dorisiana*	*S. sclarea*	*S. somalensis*
EO (Y)					
*S. dolomitica*		-	1.78 (0.56–6.02)	0.87 (0.29–2.63)	0.25(0.09–0.61)
*S. dorisiana*		0.56 (0.17–1.80)	-	0.49 (0.13–1.80)	0.14(0.04–0.45)
*S. sclarea*		1.14 (0.38–3.50)	2.04 (0.55–7.97)	-	0.29(0.09–0.83)
*S. somalensis*		4.00 (1.63–10.68)	7.07 (2.22–26.29)	3.46 (1.21–10.87)	-

Relative median potency analyses (rmp) values of probits (EO in column vs. EO in row): Values < 1 indicate higher repellence, values > 1 indicate lower repellence. Bold indicates significant values (95% CI ≠ 1).

**Table 6 biology-09-00206-t006:** Complete protection time (CTP) of *Salvia dorisiana*, *S. dolomitica*, *S. sclarea,* and *S. somalensis* essential oils (EOs) against *Aedes albopictus* females.

EO.	CPT ^a^
*S. dolomitica*	21.45 ± 7.12 ^ab^
*S. dorisiana*	43.28 ± 3.43 ^a^
*S. sclarea*	13.60 ± 2.31 ^ab^
*S. somalensis*	4.60 ± 2.70 ^b^

^a^, Complete protection time values (min) of *Salvia* EOs applied on human skin at the dose of 20 nL cm^−2^; different letters indicate significant differences among the same dose of each EO (Kruskas–Wallis, Dunn–Bonferroni pairwise comparisons, *P* ≤ 0.05).

**Table 7 biology-09-00206-t007:** Main odors that characterized the smell of *Salvia dolomitica*, *S. dorisiana*, *S. sclarea*, and *S. somalensis* essential oils.

	Species	*S. dolomitica*	*S. dorisiana*	*S. sclarea*	*S. somalensis*
Odour Class					
Vegetative odours		Herbaceous	Herbaceous	Citronella	Herbaceous
		-	Mint	Fresh mint	Menthol
		-	-	Citrus	Chamomille
		-	-	Lime	-
Spicy		-	Sandalwood	Green spicy	Green spicy
		-	Licorice	Thyme	Thyme
		-	-	Sage	Green tea
Other		-	Resin	-	-
Off-flavors		Mould	-	-	-
		Wet rag	-	-	-
		Old soap	-	-	-
		Petrol	-	-	-

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
