# Peer review of "Salvia Spp. Essential Oils against the Arboviruses Vector Aedes albopictus (Diptera: Culicidae): Bioactivity, Composition, and Sensorial Profile—Stage 1"

_biology, 2020, doi:10.3390/biology9080206_

Round 1
Reviewer 1 Report
The paper describes the potential of Salvia essential oils as repellents and biopesticides. Additionally, it also adds the sensory evaluation of the tested oils. Althought the scientific information is not super novel and suprising, I think it deserves to be publish in the journal.
Actually, I have nothing much to criticise about the design, execution and evaluation of the study; the statistics are sound, the tables clear and understandable.
I just wonder, do authors feel, that the efficacy test method used for cpt calculation could not be affected by decreasing activity of mosquitoes? The statement says: "To verify the mosquitoes’ readiness to bite, the control and the EOs treated hand were regularly interchanged during each test. A new mosquito’s cage was used when no mosquito attempted to bite the untreated hand." Does this mean, that if even one out of all the mosquitoes in the cage was probing, the test continued? Does this ever happened?
I also feel the english sometimes need a revision (e.g L40-41 should read: Mosquitoes (Diptera: Culicidae) belong among the most serious threats for human beings because of their ability to transmit viruses and parasites.) I.
Similarly, few minor formal flaws were found throughout the text. (e.g. the sentences should not start with abbreviation as it is at L153: "S. sclarea exhibited the..")
Author Response
Reply to Reviewer 1
Many thanks for the time you spent to correct our manuscript. Your corrections and recommendations have been totally performed and have certainly improved the quality of the text.
With regard to Figure 5, the Reviewer is right: there was a mistake in the legend of the figure and we corrected it.

Reviewer 2 Report
Please see attached file for your reference

Author Response
All comments have been addressed.

Reviewer 3 Report
Article: Salvia spp. essential oils against the arboviruses vector Aedes albopictus (Diptera: Culicidae): Bioactivity, composition, and sensorial profile – Stage 1
General notes:
The essential oils of the same two species of Salvia: dorisiana, S. sclarea were previously studied for the same aim for their repellent activity against Aedes albopictus. In addition, the compositions of their essential oils were also investigated [Conti et al. 2012]. Why the both species repeated in this wok with same aims.
Lines numbers:
Line 31: “Ae. Albopictus”; abbreviation of genus name better to write with one letter as (A. Albopictus).
Line 171: table 1. “Oxygenated Diterpenes”, these are reported only in “S. sclarea”. Generally, are diterpenes found as components of EOs.
Line 253: what is the difference between “insecticides” and “insect repellents”. Which term is suitable for essential oils.
Line 257: “In this work, we observed that the Salvia EOs showed a wide range of toxicity…”. Do you mean all species of Salvia or only the tested species of Salvia in this work.
Author Response
All comments have been addressed.

Reviewer 4 Report
The authors are to be congratulated for an interesting a readable item of research. There appear to be few errors here, excepting some minor linguistic alterations outlined below. The topic is of interest to the readers of Biology and the manuscript is written with a clear logical progression. Recommended for publication.
Notes:
Introduction
[grammar correction]
41: Mosquitoes (Diptera: Culicidae), are among the most serious threats for human beings because of
42 their ability to transmitting viruses and parasites.
[grammar correction]
108 products on the day of the bioassay. All volunteers were informed about the experiment and provided
109 their written consent.
Discussion
[grammar correction: note singular use of agreement not pleural agreements]
264 characterized by GC-EIMS. The EO profiles of the studied species were completely in agreements
[introduce space after [34]]
266 co-worker [34] on clary
Author Response
All comments have been addressed.

Reviewer 5 Report
The manuscript entitled “Salvia spp. essential oils against the arboviruses vector Aedes albopictus (Diptera: Culicidae): Bioactivity, composition, and sensorial profile – Stage 1” combine the chemical and sensorial profiles with the bioactivity data of Salvia spp. essential oils to select the most suitable to be used as repellent and insecticide against the invasive mosquito Aedes albopictus (Diptera: Culicidae). In my opinion, this manuscript is suitable to be publish in Foods after revisions.
The authors should standardize the units through the manuscript, sometimes appears ml other mL, and minutes other min.
Sensory analysis, based on the manuscript, this is performed by 10 assessors “expert panel”. How were these assessors trained to carry out this analysis?
Table 1. Please add the retention time and the linear retention indices reported in the literature for a similar column for each compound identified.
Author Response
All comments have been addressed.

Round 2
Reviewer 2 Report
The manuscript can be accepted.
Author Response
We would like to thank the reviewer for the time spent in revisioning the manuscript
Reviewer 3 Report
Article: (Salvia spp. essential oils against the arboviruses vector Aedes albopictus (Diptera: Culicidae): Bioactivity, composition, and sensorial profile – Stage 1)
You used in this work a method for “EOs Repellent activity”; but you speak also about insecticide activity across the manuscript; even you described the differences between the two terms in you response for the previous review. Can you improve this point.
Author Response
Reply to reviewer 3 (round 2)
You used in this work a method for “EOs Repellent activity”; but you speak also about insecticide activity across the manuscript; even you described the differences between the two terms in you response for the previous review. Can you improve this point.
R. We spoke about insecticidal activity because the EOs were tested also as larvicides (see lines 95-101). We are aware of the poor applicability of essential oil-based larvicides (and this is probably why it have been not noticed by the reviewer), but we set up the larvicidal tests for completeness of the research on the bioactivity of the Salvia EOs
Reviewer 5 Report
The manuscript entitled "Salvia spp. essential oils against the arboviruses vector Aedes albopictus (Diptera: Culicidae): Bioactivity, composition, and sensorial profile – Stage 1" should be accepted in the present form. The authors performed all corrections suggested by the reviewers.
Author Response

(The authors gave the same response as above.)
